# Histopathological Signatures of the Femoral Head in Patients with Osteonecrosis and Potential Applications in a Multi-Targeted Approach: A Pilot Study

**Giovanna Desando** [1,*,†], **Livia Roseti** [1,†], **Isabella Bartolotti** [1,*], **Dante Dallari** [2], **Cesare Stagni** [2] **and Brunella Grigolo** [1]

1    SSD Laboratorio RAMSES, IRCCS Istituto Ortopedico Rizzoli, 40136 Bologna, Italy; livia.roseti@ior.it (L.R.); brunella.grigolo@ior.it (B.G.)
2    Chirurgia Ortopedica Ricostruttiva Tecniche Innovative-Banca del Tessuto Muscoloscheletrico (BTM), IRCCS Istituto Ortopedico Rizzoli, 40136 Bologna, Italy; dante.dallari@ior.it (D.D.); cesare.stagni@ior.it (C.S.)
*    Correspondence: giovanna.desando@ior.it (G.D.); isabella.bartolotti@ior.it (I.B.); Tel.: +39-051-6366803 (G.D.); +39-051-4689945 (I.B.)
†    Giovanna Desando and Livia Roseti equally contributed to the work.

**Abstract:** (1) Background: Osteonecrosis (ON) of the femoral head is a disabling disease for which limited treatment options exist. Identifying therapeutic targets of its evolution could provide crucial insights into multi-targeted approaches. The aim of this pilot study was to assess the histopathological features of patients with non-traumatic femoral head (NTFH) and post-traumatic femoral head (PTFH) ON to produce a fresh vision for clinical use. (2) Methods: We got biopsies from patients with different ON stages, according to the ARCO system. Samples from multi-organ donors were used as controls. Histological and immunohistochemical evaluations were performed on the osteochondral unit. (3) Results: The PTFH group displayed several fibrotic reactions, a small stem cell pool and a lower international cartilage repair society (ICRS)-I score than NTFH, which instead presented intact cartilage similar to the controls. Immunostaining for collagen I and autotaxin confirmed these features in the PTFH group, which displayed top levels of MMP-13 involved in cartilage loss and reduced CB-2 in the underlying bone. Both groups manifested a similar pattern of apoptotic and pain mediators. (4) Conclusions: The different histopathological features suggest a multi-disciplinary and multi-targeted approach for ON. Further studies are necessary to measure the effect size to gain clinical evidence.

**Keywords:** osteonecrosis; osteochondral unit; tissue remodelling and repair; multi-targeted approach

## 1. Introduction

Osteonecrosis (ON) of the femoral head is a progressive and disabling disease, affecting active patients between the third and fifth decade of life with a high burden on the healthcare system [1–3]. The pathogenesis of ON involves the interplay of genetic, local and metabolic aspects with a different incidence rate among men and women (ratio male/female: 4:1) [4–7]. It is possible to identify two major types of ON aetiologies. The aetiology in ON patients with post-traumatic femoral head (PTFH) ON includes previous traumatic events; non-traumatic femoral head (NTFH) ON aetiology includes corticosteroid use, alcohol abuse, obesity, autoimmune diseases, and immunosuppressive therapies [7–9]. The pathology reflects a dynamic course leading to femoral head collapse because of subchondral bone fractures and inadequate bone repair [10–12]. Skeletal complications and pain in

ON patients occurs because of abnormal osteoblastic and osteolytic activities [13,14]. In particular, ON patients showed impaired action of the OPG/RANKL/RANK signalling pathway [9,14]. Wang X. and his group observed a similar behaviour whereby ON patients displayed increased levels of osteoprotegerin (OPG), receptor activator of nuclear factor-kB (RANK), and its ligand (RANKL) genes in ON necrotic areas [15]. Beyond such classical signalling pathways, scientists have even started testing the role of cannabinoids (CN) in bone repair, as bone-marrow-derived osteoclasts and osteoblasts [16], MC3T3 E1 osteoblast-like cells [17] and osteocytes express CN receptors [18]. Beyond the critical role of bone in the ON setting, scientists have pointed great attention towards the articular cartilage, because of its close anatomic continuity and communication with the underlying subchondral bone [19,20]. In particular, several authors stressed the clinical significance of considering structural cartilage changes following the mechanical stress induced by the collapse of the subchondral bone in ON [21–23]. However, these changes contribute to altering the biomechanics of the joint leading to OA progression and finally to joint destruction.

There is no gold-standard treatment for ON because of the controversial results [24,25]. Selecting treatment options for ON depends on many factors, including the pathology stage, patients' age and health conditions, and lesion size and location [26,27]. In particular, treatments for ON management in the pre-collapse stage include non-surgical (weight control), pharmacological or biophysical techniques, and mesenchymal stromal cell (MSC)-based therapy [27–31]. Total hip arthroplasty is among the best therapeutic alternatives after the collapse of the femoral head. However, it has its disadvantages (e.g., infection, revision, and dislocation) [24,25]. Therefore, many scientists focused on femoral head regeneration. In 2002, Hernigou P. and his group reported promising results following MSC therapy in ON patients in the pre-collapse stage at 5 to 10 years of clinical follow-up [29]. A recent review summarised that MSCs could regenerate the necrotic area of the femoral head by injecting the suspension into the lateral artery of the circumflex or loading on carriers via core decompression and implantation [32]. Testing the crosstalk between cartilage and the underlying subchondral bone is essential to achieve global comprehension of physiological responses in ON disease. Differences between ON patients with various aetiologies could give first perspectives for tailored-based therapies. In this light, we conducted a small-scale preliminary (pilot) study aimed at evaluating histopathological features in two ON patient groups with NTFH and PTFH. Study design foresaw investigating several mediators modulating the osteochondral unit to get more knowledge of potential therapeutic targets for ON. In particular, we selected collagen I and autotaxin (ATX) to test the fibrotic reactions commonly causing poor mechanical properties and the limited capacity of MSCs to differentiate towards collagen type II in cartilage [33,34]. The axis ATX/lysophosphatidic acid (LPA) regulates collagen type I biosynthesis and plays essential functions in bone metabolism, thus resulting in an attractive molecular target [34,35]. We tested tissue destruction by selecting matrix metalloproteinases (MMP-13) and aggrecanases (ADAMTS-5), causing the proteolytic cleavage of collagens and the aggrecan protein [36]. As for apoptosis reactions, we tested active caspase 3, which is a well-known biochemical marker of both early- and late-stage apoptosis [37]. We tested the endocannabinoid receptor-2 (CB-2), as it is present in distinct cell types of the joint, like chondrocytes, bone cells, progenitor cells during osteoarthritis and rheumatoid arthritis with potential therapeutic implications [38]. We chose the neurotrophin nerve growth factor (NGF) and the nociceptive peptide substance P (SP) to test the pain response [39,40].

## 2. Materials and Methods

### 2.1. Patient Data and Surgical Procedure

This study obtained approval from the Ethics Committee of IRCCS Istituto Ortopedico Rizzoli (Prot. gen. n. 26146 del. 31.10.2006). Eleven male patients with clinical and radiological signs of ON of the femoral head gave their informed written consent to this study. We selected patients according to specific inclusion and exclusion criteria. Inclusion criteria foresaw the enrolment of

male patients with a mean age between 18–50 years and clear signs of hip osteonecrosis by MRI. Exclusion criteria foresaw the exclusion of patients showing metabolic diseases, rheumatoid arthritis, autoimmune and neurological disorders. In this study, five ON patients showed NTFH (mean age 37 ± 4; range: 30–43), whereas six ON patients had PTFH (mean age/SD: 30 ± 3; range: 25–37). The aetiology of the NTFH group included glucocorticoid treatment. ON patients included in the NTFH group did not report aetiology for alcohol abuse and autoimmune diseases; only one patient had a chronic bowel disorder. NTFH and PTFH patients underwent preoperative MRI and X-ray using the Association Research Circulation Osseous (ARCO) evaluation system [41]. This system considers the size of the necrotic lesion, its femoral head extent, and the joint involvement (Stage I: normal on X-ray and CT; Stage IV: the destruction of joint with secondary arthritic changes; A, B and C describe the extent of ON involvement: A: non-articular, B: medial; C: central). We reported a scientific diagram of the weight-bearing area of the femoral epiphysis where we harvested biopsies (see Supplementary Figure S1). Patients' femoral heads underwent surgical repair with a synthetic resorbable osteochondral scaffold plug. During the procedure, surgeons collected osteochondral biopsies of the lesions from the weight-bearing area of the femoral head with a 10-mm diameter through a 12–14-mm deep trocar. All samples were processed for histological and immunohistochemical analyses. We used the femoral head from three male multi-organ donors (mean age/SD: 35 ± 5; range: 30–40) as healthy controls. Multi-organ donors did not suffer from any musculoskeletal disease. We selected the donors through the bone bank program for tissue donation after the family' s donor consent. Femur harvesting was performed within six hours from asystole, and involved its excision and placement in Dulbecco-modified Eagle medium with L-glutamine, $NaHCO_3$, and antibiotics, and storage at 4 °C.

### 2.2. Histological Assessment

Osteochondral samples were fixed with 10% buffered formalin and decalcified in 4% hydrochloric acid and 5% formic acid [42]. After processing with a graded alcohol series, specimens were embedded in paraffin. We tested proteoglycan and collagen content by staining tissue sections with 0.1% Safranin-O/0.02% Fast Green (Sigma Aldrich, St Louis, MO, USA). International Cartilage Repair Society (ICRS)-I score was used for evaluating the histological features [43]. This score considers six parameters: surface, matrix organization, cell distribution, cell viability, subchondral bone and cartilage mineralization. It has a range from 0 (presence of fibrous tissue) up to 18 (presence of healthy osteochondral tissue). We assessed necrotic bone lesions with the Ficat and Arlet classification system [44]. This system considers four types of bone necrosis. Its score ranges from 0 (slight disease) to 4 (severe disease). Six microscopic fields, spaced 20 sections, were assessed for each sample by two blinded investigators (GD, IB) with an Eclipse 90i microscope (Nikon, Melville, NY, USA).

### 2.3. Immunohistochemical Analyses

Analyses for collagen type I, caspase-3, MMP-13, ADAMTS-5, autotaxin, NGF, SP, and CB-2 were performed. After antigen retrieval with 0.1% proteinase (Sigma) at 37 °C for 20 min, the sections were blocked with 2% bovine serum albumin (Sigma) in phosphate-buffered saline for 30 min. Then, an incubation with human collagen type I (2 µg/mL; Chemicon International, Temecula, CA, USA), caspase-3 (5 µg/mL; R&D Systems), ADAMTS-5 (1 µg/mL, Abcam), MMP-13 (5 µg/mL, R&D Systems, Minneapolis, MN, USA), autotaxin (2 µg/mL; R&D Systems), NGF (1 µg/mL, Chemicon), and CB-2 (5 µg/mL; Novus Biologicals) was carried out. Specific negative controls were performed by omitting the primary antibodies or using an isotype-matched control while we stained nuclei with CAT hematoxylin (Biocare Medical). Six microscopic fields (100× magnification) were assessed for each sample by a blinded investigator with a semi-quantitative method. We firstly segmented cartilage and the subchondral bone for each marker by selecting zones apart from the tidemark, especially for the PTFH group where the tidemark was fragmented. Image acquisition and processing with an Eclipse 90i microscope (Nikon) and NIS-Elements Software were used for the image analysis of stained sections

with the Hue/Saturation/Intensity (HSI) system. Hue (H) was assessed by setting the threshold for positive pixels at 220 to 255. Ranges from 0 to 150 were threshold values for S and I. The measurement of positive cells and area for each marker was done on the entire osteochondral sample (10× objective lens) and expressed as a percentage of positive cells and area on a scale from 0 (no protein expression) to 100 (the highest protein expression).

### 2.4. Statistical Analysis

Graph Pad Prism software was adopted for statistical analysis. The Kolmogorov–Smirnov test was used to test the data distribution. We used the Mann–Whitney U test for unpaired data to assess differences in NTFH and PTFH groups. We reported data in a scatter plot graph with mean ± standard deviation (SD). $p < 0.05$ was considered significant.

## 3. Results

### 3.1. Radiographic Assessment

According to the ARCO osteonecrosis classification system, patients with NTFH and PTFH displayed different stages of ON. The NTFH group reported three patients with III C stage and two patients with IV C. Patients with III C stages showed clear signs of ON and separation of the subchondral bone from the necrotic cancellous bone. Patients with IV C revealed joint space narrowing following the femoral head collapse. The PTFH group included four patients with IV C and two with IV B, which reported bone fracture and subsequent arthritic changes.

### 3.2. NTFH and PTFH Groups Displayed Different Histological Features in Cartilage and Bone

The control group showed a regular cartilage surface, adequate cell distribution and rich proteoglycan content. Bone tissue was also well-structured with osteocytes embedded in the bone matrix and trabecular spaces containing bone marrow and blood vessels (Figure 1a). The NTFH group showed a regular cartilage surface with small discontinuities and good proteoglycan content. The extracellular matrix of specimens with radiographic IV C stage displayed a reduced number of cells, some cell clones, and tidemark discontinuities. Bone tissue showed a low number of osteocytes and necrosis of the bone marrow in the trabecular spaces. No inflammatory reactions were, however, present (Figure 1a). The PTFH group exhibited a typical fibrocartilaginous aspect with several cracks in the superficial zone, and an altered cell arrangement with round cells interposed in the extracellular matrix. The tidemark displayed non-continuous areas with cells migrating from the subchondral bone towards the cartilage. The PTFH group showed trabecular spaces containing fibroblasts, blood vessels, and osteoclasts but no inflammatory infiltrate (Figure 1a). The NTFH group showed a higher ICRS-I score than the PTFH group ($p < 0.05$), reporting mean values of 13.1 ± 0.6 and 7.8 ± 0.9, respectively (Figure 1b). The PTFH group exhibited a worse histological aspect, different from the control group, which reported a mean value of 16.8 ± 0.6 ($p < 0.001$) (Figure 1b). As for the cartilage parameters, the extracellular matrix and the tidemark showed better organization in NTFH rather than in the PTFH group ($p < 0.001$) (data not shown). The Ficat classification system gave evidence of more degenerative changes in the subchondral bone marrow of PTFH group when compared with NTFH group but with no statistical evidence (Figure 1b).

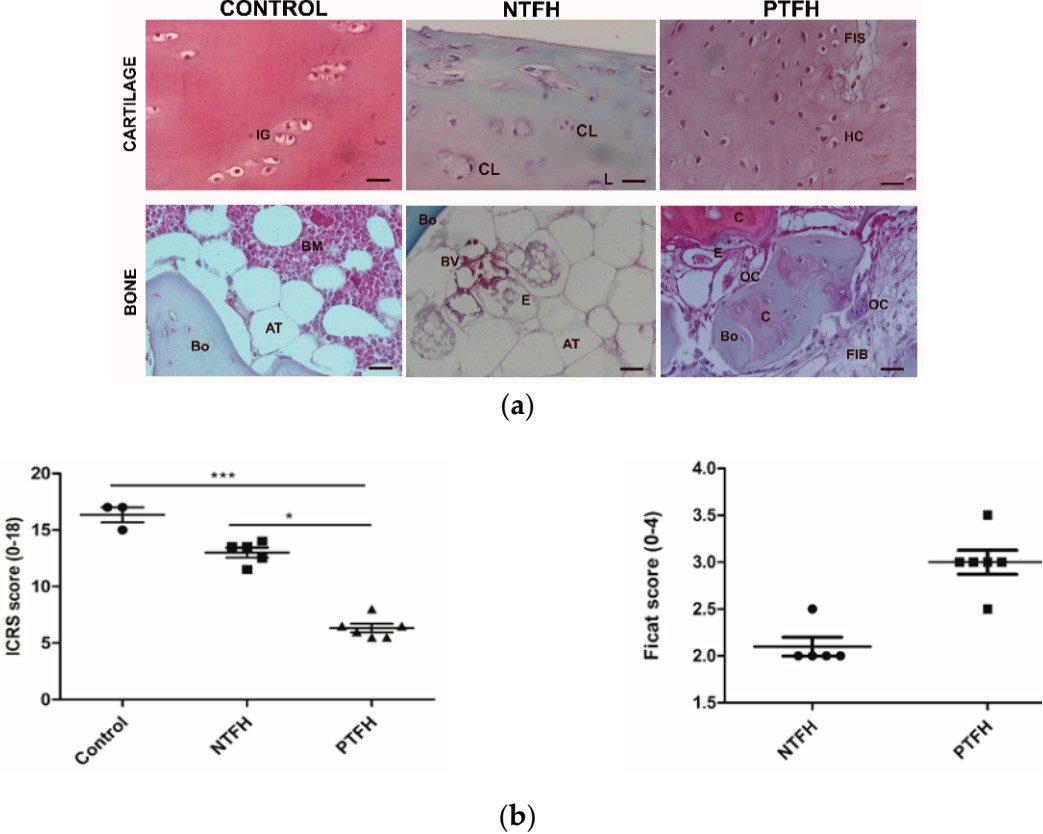

**Figure 1.** (**a**) Safranin-O/Fast Green staining of cartilage and subchondral bone from healthy controls, and two groups of patients with non-traumatic femoral head (NTFH) ON (*n* = 5), and post-traumatic femoral head (PTFH) ON (*n* = 6). Red: proteoglycans; green: collagen. Scale bar: 50 µm. IG isogenic groups: CL cell clusters; L: empty lacunae; FIS fibrillation processes; HC hypertrophic chondrocytes; T tears; Bo bone tissue; BM bone marrow; AT adipose tissue; BV blood vessels; FIB fibrous tissue; OC osteoclasts; E erythrocytes. (**b**) Graphical representation of ICRS and Ficat scores of the osteochondral unit of control, NTFH and PTFH groups. Data are reported in a scatter plot graph with mean ± standard deviation (SD). * *p* < 0.05: NTFH versus PTFH; *** *p* < 0.001: PTFH versus the control group.

### 3.3. PTFH Group Displayed a Higher Expression of Fibrotic Markers than NTFH Specimens

The control group showed low protein expression for type I collagen in cartilage, whereas we noticed a high percentage of this marker in the underlying subchondral bone (Figure 2a,b). Fibrosis reactions in terms of the presence of type I collagen were more robust in the cartilage of the PTFH group compared to the NTFH group (Figure 2a,b). This latter group showed higher type I collagen expression compared to the control (*p* < 0.05) and NTFH groups (*p* < 0.05). As for autotaxin, the two ON groups did not show any difference; however, the PTFH group reported a higher amount of this marker compared to the control group (*p* < 0.05). The NTFH group showed mild protein expression for both collagen type I and autotaxin, especially in the articular cartilage.

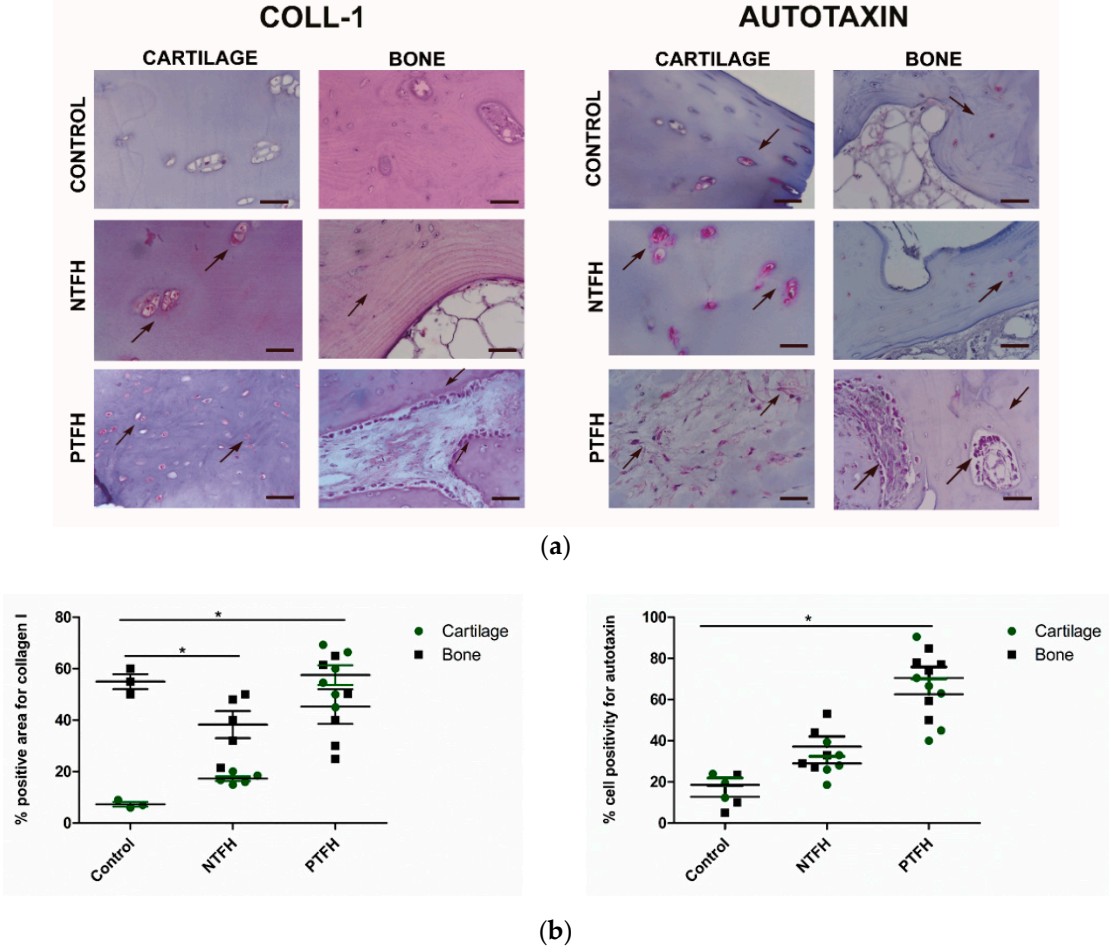

**Figure 2.** (**a**) Immunostaining for collagen type I and autotaxin of control, NTFH and PTFH groups. Scale bar: 50 μm. Black arrows: positive areas for markers. (**b**) Graphical representation of quantitative measurements for collagen type I and autotaxin in control, NTFH and PTFH groups. Data are reported in a scatter plot graph with mean ± SD. Collagen type I: * $p < 0.05$: Control versus PTFH group; * $p < 0.05$ NTFH versus PTNH group. Autotaxin: * $p < 0.05$: Control versus PTFH group.

We did not find any difference for caspase-3 between the NTFH and PTFH groups. Both specimens displayed higher cell positivity for this marker when compared to the control group (Figure 3a,b). Regarding CB-2, a molecule involved in bone remodelling and pain responses, the control group displayed moderate expression, especially near bone marrow spaces, and at a lesser extent in the osteocytes. The NTFH group revealed higher protein expression for CB-2 in chondrocytes than in osteocytes and bone marrow precursors within the subchondral bone. The PTFH group displayed a similar behaviour (Figure 3a,b).

To assess the breakdown of the extracellular matrix, we analysed ADAMTS-5 and MMP-13 as catabolic markers. All cartilage and bone specimens showed low levels of ADAMTS-5, with no difference between the two ON groups (Figure 4a,b). As for MMP-13, the PTFH group displayed a higher cell positivity in the middle and deep layers of articular cartilage when compared to the control and NTFH groups ($p < 0.05$) (Figure 4a,b).

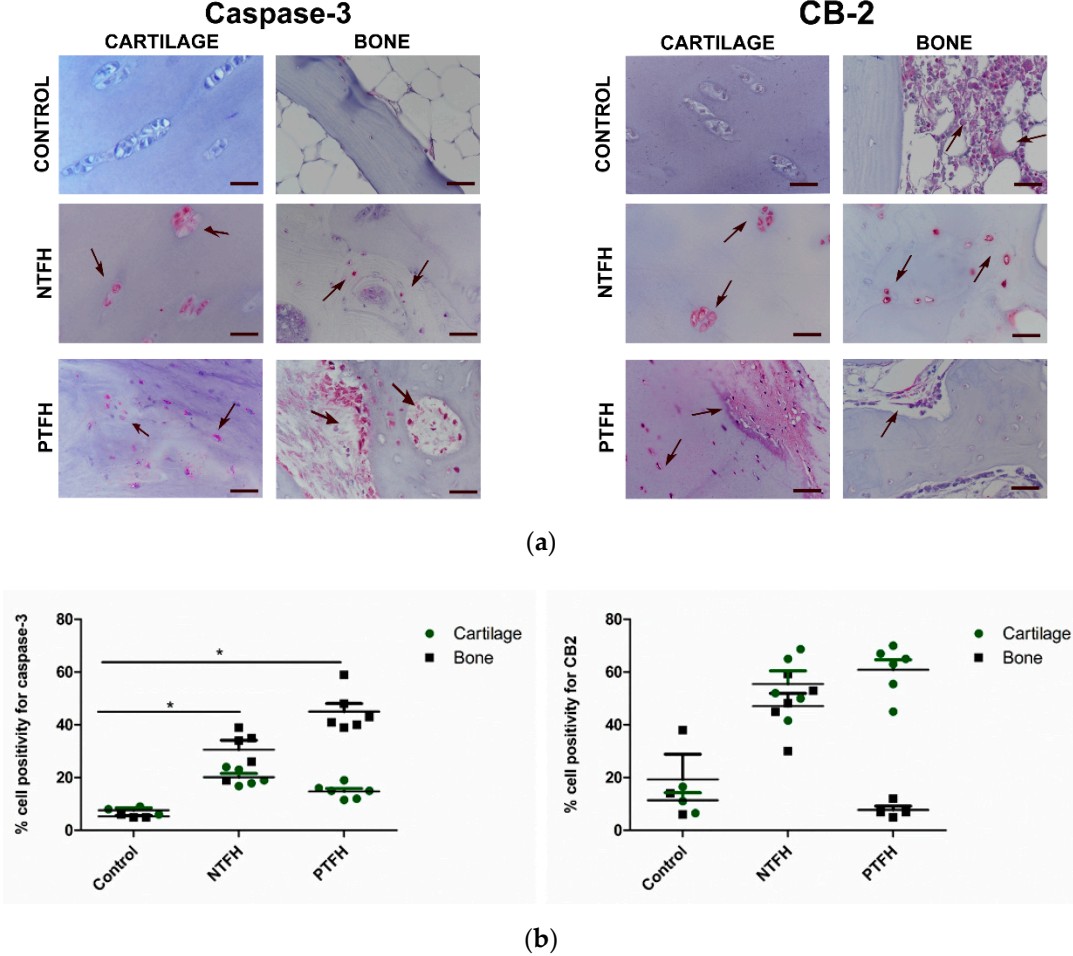

(a)

(b)

**Figure 3.** (**a**) Immunostaining for caspase-3 and CB-2 of the control, NTFH and PTFH groups. Scale bar = 50 μm. Black arrows show positive cells. (**b**) Graphical representations of the percentage of positivity for caspase 3 and CB-2. Data are expressed in a scatter plot graph with mean ± standard deviation (SD). Caspase 3: * $p < 0.05$: Control versus NTFH group; * $p < 0.05$: NTFH versus PTFH group.

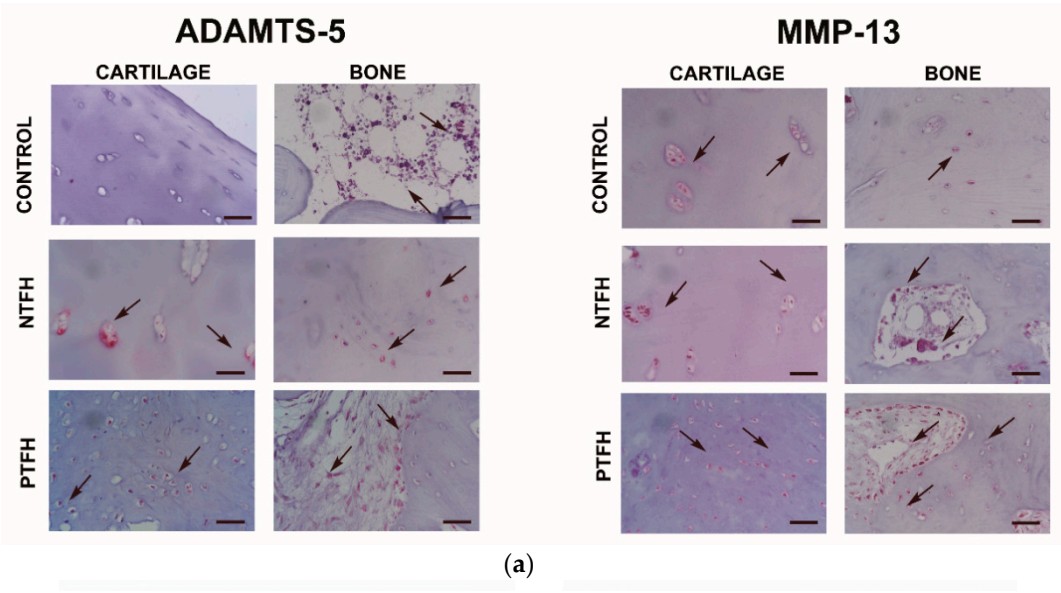

(a)

**Figure 4.** *Cont.*

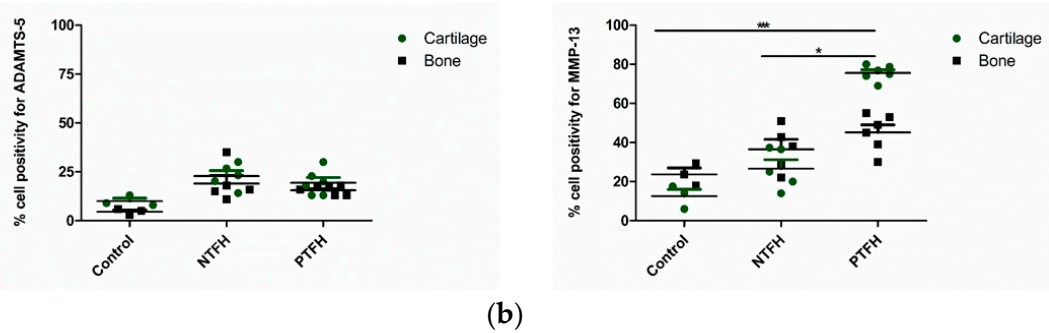

**(b)**

**Figure 4.** (**a**) Immunostaining for ADAMTS-5 and MMP-13 in osteochondral tissues from the control, NTFH and PTFH groups. Scale bar = 50 μm. Black arrows show positive cells. (**b**) Graphical representations of the percentage of positivity for ADAMTS-5 and MMP-13. Data are expressed in a scatter plot graph with mean ± standard deviation (SD). MMP-13: ** $p < 0.001$: Control versus PTFH group; * $p < 0.05$: NTFH versus PTFH group.

We analysed NGF and the substance P (SP) nociceptive fibre pattern to test the pain response. Both NTFH and PTFH groups showed higher immunostaining for NGF in the cartilage rather than in the underlying subchondral bone ($p < 0.05$) (Figure 5a,b). Both ON groups displayed an increased SP positivity near cell clones of the articular cartilage and hypercellular and fibrotic areas of the subchondral bone (Figure 5a,b).

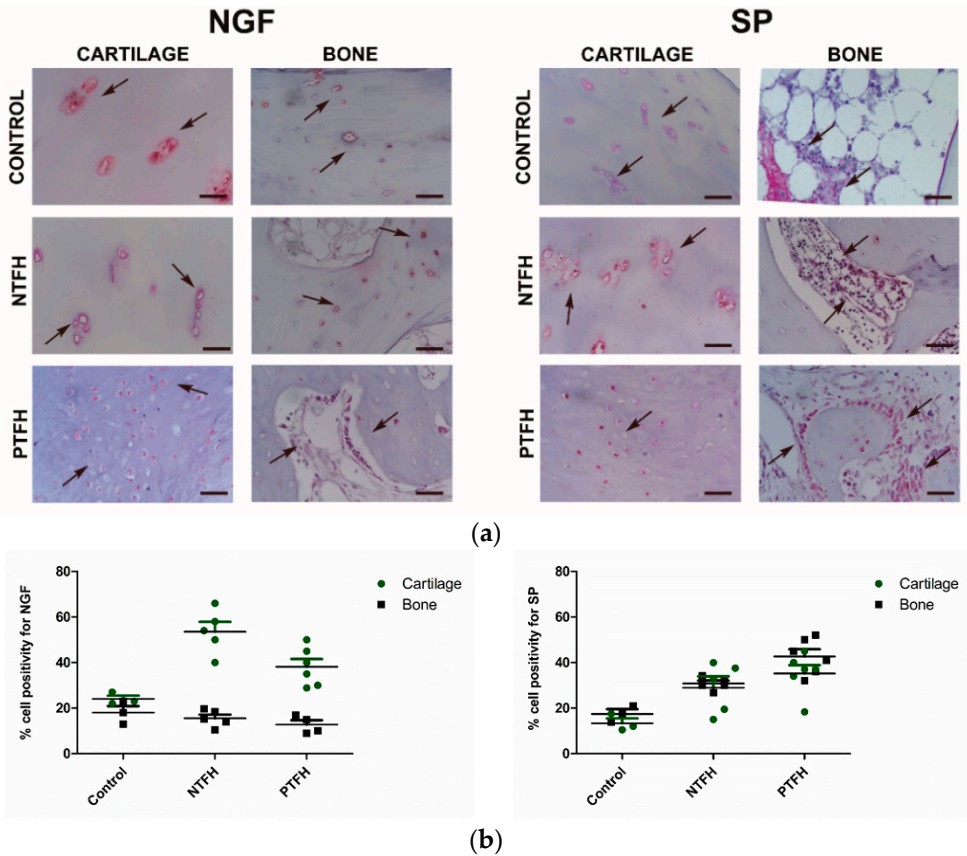

**Figure 5.** (**a**) Immunostaining for NGF and SP in osteochondral tissues from control, NTFH and PTFH groups. Scale bar: 50 μm. Black arrows: positive areas for the selected markers. (**b**) Graphical representation of quantitative measurements for NGF and SP in control, NTFH and PTFH groups. Data are expressed in a scatter plot graph with mean ± standard deviation (SD).

## 4. Discussion

The search for ON therapies capable of controlling the joint microenvironment is a unique challenge [1,2]. A holistic view of biological responses during ON can produce insights for generating efficient strategies [22]. Our pilot study showed different histopathological features in two groups of ON patients by opening up interesting biological perspectives. All specimens with NTFH and PTFH forms displayed basal integration with the underlying bone. However, PTFH samples showed noticeable fibrotic reactions and a reduced stem cell pool in the bone marrow. Interrupting blood supply likely contributed to mediating the death of bone and marrow cells. As the bone supports the exchange of nutrients with cartilage by facilitating joint force distribution, bone alterations might speed up cartilage degeneration [22,45]. The subchondral bone changes may lead to misalignment of the articulating surfaces with consequences on the mechanosensory cells in the bone [45]. Thus, first-line approaches for shifting the bone matrix turn-over are essential to avoid progressive degenerative changes in cartilage tissue. Bisphosphonate treatment could restore the balance between bone resorption and formation [26]. Biopsies from patients with post-traumatic aetiology reported several OA features, including impaired extracellular matrix and vascular infiltration in cartilage because of the mechanical stresses. We determined the relevance of matrix composition and tidemark presence using a semi-quantitative analysis with the ICRS score, by observing different histological scores in the PTFH and NTFH groups. This latter displayed similar results to the control group by reporting intact cartilage but several changes in the subchondral bone. These findings are in line with other studies, which showed intact articular cartilage in the NTFH group with corticosteroid treatments [22]. The two groups even exhibited distinct protein expression for mediators modulating fibrotic and catabolic responses.

Protein assessment for collagen type I and autotaxin corroborated the fibrotic aspect in both cartilage and bone from the PTFH group. The autotaxin–lysophosphatidic acid (LPA) axis is emerging as a critical regulator in various biological responses [46]. Autotaxin exerts a fibrotic activity by catalysing LPA, which promotes fibrosis responses by regulating collagen type I biosynthesis [34,47]. Moreover, autotaxin controls the fusion and bone resorption capacity of osteoclasts [35]. Another peculiar feature observed in the cartilage and bone tissues from the two ON aetiologies was the protein expression for MMP-13. It is a molecule which stimulates collagen and proteoglycan degradation in cartilage and bone [48]. Along this path, Grassel S. et al. provided first evidence of MMPs in the ON setting. ON patients with PTFH showed increased gene expression levels of MMP-2 and a low amount of the tissue inhibitor of metalloproteinases (TIMPs) [49]. The increased protein levels of MMP-13 in the PTFH group may likely depend on the up-regulated activity of chondrocytes in producing MMPs after stress forces. These findings open preliminary biological insights into considering MMPs and fibrotic markers as attractive therapeutic targets by exploiting future technological advances like CRISPR-Cas9 [50]. Combining current ON strategies with small molecule inhibitors to block collagen type I, MMP-13 and ATX could improve the success of PTFH treatment [47,51]. Regardless of the ON aetiology, samples from both groups displayed a moderate positivity for caspase-3, showing higher values than the control. Beyond the classical signalling pathway, we considered the role of CB-2 involved in the endocannabinoid system. Jiang S. et al. reported that this system promotes specific signalling pathways in response to pathogenic events to launch repair processes [52]. In specimens from both groups, we showed positive immunostaining for CB-2, especially in cartilage, with higher values than healthy controls. We noticed a low expression of CB-2 in the bone marrow niche, likely because of the small number of osteoprogenitors in ON patients.

Like in ON disease, femoral heads from patients with the atrophic form of osteoarthritis (OA) display bone marrow lesions with numeric, topographic and functional variations of MSCs [53,54]. Bone marrow perturbations reduce the bone repair and remodelling activities of MSCs and lead to damage of the overlying articular cartilage [55,56]. Targeting both cartilage and bone turn-over and the crosstalk between their cell types may be a valuable approach [56]. In this light, there is abundant evidence of the powerful effects of MSCs for promoting osteochondral repair and inhibiting

inflammatory and fibrotic reactions in the OA environment [57–59]. Optimal therapeutic approaches might envisage several interventions, at varying stages of OA and ON disorders and selecting specific patient features. Restoring the stem cell pool, especially in the PTFH group, through cell-based therapy, could be a feasible intervention. The remarkable bone–cartilage interface imbalance of this patients group, simultaneously with the low stem cell pool, may benefit from the differentiation and paracrine properties of MSCs. Hernigou P et al. reported promising clinical results using autologous bone marrow transplantation in ON cases by highlighting the biological and therapeutic value of this strategy [30]. Along the way, several clinicians have observed the combination of autologous bone marrow concentrate with core decompression contribute in a significant manner to decelerating ON progression by limiting total hip arthroplasty [60,61].

Finally, this study considered pain implications in the biopsy samples from two ON patients focusing on NGF and substance P; this latter is involved in the onset of inflammatory processes and pain transmission [52]. Specimens from both groups manifested pronounced immunostaining for NGF in cartilage. Beyond its role as a neurotrophic factor, NGF impairs the migratory and matrix remodelling activities of cartilage and stem progenitor cells [62,63]. More in-depth investigations are crucial to assess the effect size and produce clinical evidence, as the small number of ON cases and heterogeneity are the major limitations of the present study. However, this pilot study gave indications to enhance the benefit of current ON approaches by merging themselves with fibrotic inhibitors and choosing a multi-disciplinary and targeted strategy to both cartilage and bone in the PTFH group.

## 5. Conclusions

Sustainable multi-disciplinary strategies can represent valid tools to tackle complex pathologies, with tremendous impact in the clinical decision-making framework. In our pilot study, the different histopathological features of NTFH and PTFH groups would suggest a multi-disciplinary and multi-targeted approach for both cartilage and bone tissues. Hypothetically, restoring the stem cell pool in the subchondral bone from the PTFH group could be fruitful in supporting tissue regeneration. Successful clinical results could be obtained by hindering the fibrotic and catabolic responses at the level of cartilage and bone in the PTFH. Further studies are necessary to measure the effect of size to gain clinical evidence.

**Supplementary Materials:** The following are available online at http://www.mdpi.com/2076-3417/10/11/3945/s1, Figure S1: Scientific diagram.

**Author Contributions:** Conceptualisation, L.R. and G.D.; methodology, G.D.; C.S. and I.B.; software, G.D.; formal analysis, G.D. and D.D.; investigation, L.R., D.D. and C.S.; data curation, G.D., C.S. and I.B.; writing-original draft preparation, L.R. and G.D.; writing-review and editing, G.D., L.R., I.B., C.S., D.D., and B.G.; supervision, B.G.; project administration, B.G.; funding acquisition, B.G. All authors have read and agreed to the published version of the manuscript.

**Funding:** This research was funded by the Italian Ministry of Health, 5× 1000 Funds anno 2016. "Malattie osteoarticolari: fisiopatologia e strategie terapeutiche innovative".

**Acknowledgments:** The authors wish to thank Patrizia Rappini and Martina Rocchi for their technical and scientific help.

**Conflicts of Interest:** The authors declare no conflict of interest. The funders had no role in the design of the study; in the collection, analyses, or interpretation of data; in the writing of the manuscript; or in the decision to publish the results.

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
