# Peer review of "Histopathological Signatures of the Femoral Head in Patients with Osteonecrosis and Potential Applications in a Multi-Targeted Approach: A Pilot Study"

_applsci, doi:10.3390/app10113945_

Round 1

Reviewer 1 Report

Giovanna Desando et al. present a histopathological study of human femoral neck osteonecrosis. They compared 5 non-traumatic femoral head necrosis patients' biopsy samples with 6 cases of post-traumatic femoral head osteonecrosis and samples obtained from multi-organ donors. They describe interesting differences in the histopathological picture, and distinct expression of several immunohistochemical markers, like collagen type I, MMP-13 or CB2. They also report a different matrix composition in the two disease entities, noticeable fibrosis in PTNFH, with the decrease of the stem cell pool.

These are original findings, with some implication in therapy, worthwhile to be published. However, several issues should be improved, as follows:

  • one weakness of the study is the small sample size. Non-traumatic osteonecrosis of the femural head is estimated to be quite frequent according to Zhao et al. 
  • there is an important concern related to the grouping of these entities: I am not convinced that NTFH with very different ethiologies (alcohol abuse, autoimmunity, steroid treatment etc.) represent a homogenous class. The authors mention: "The aetiology of the osteochondral femoral head defect among the NTFH group included glucocorticoids treatment." Were autoimmune and other systemic diseases excluded? Please specify more the ethiology.
  • Did the multi-organ donors suffer from any muskuloskeletal disease? This is to mention in Materials and Methods.
  • it would be of interest some systematization of the manuscript. For example, a short presentation and reasoning of the markers studied, in the section Introduction.
  • at the description of immunohistochemical analysis, the authors mention: "The measurement of positive cells and area for each marker was done on the entire osteochondral sample (10× objective lens) and expressed as a percentage of positive cells and area on a scale from 0 (no protein expression) to 100 (the highest protein expression)." How many fields did they measure in average for each sample? Please specify.
  • there is no description, how the staining intensity was assessed in cartilage and subchondral bone. Did they select zones apart from the  tidemark, if this was fragmented? Please specify this.   

There is an overall scientific merit of this study, and, with these improvements, the manuscript gains the quality to be published.

Author Response

Responses to reviewer 1

We thank Reviewer 1 and followed his/ her suggestions to improve the quality of the manuscript.

  • One weakness of the study is the small sample size. Non-traumatic osteonecrosis of the femural head is estimated to be quite frequent according to Zhao et al. 

-) We agree with the reviewer. We reported the small sample size of ON groups as a major limitation in the discussion on page 12 in lines 311-313 and also in the conclusions on page 13 in lines 325-326.

  • There is an important concern related to the grouping of these entities: I am not convinced that NTFH with very different ethiologies (alcohol abuse, autoimmunity, steroid treatment etc.) represent a homogenous class. The authors mention: “The aetiology of the osteochondral femoral head defect among the NTFH group included glucocorticoids treatment.” Were autoimmune and other systemic diseases excluded? Please specify more the ethiology.

-) We agree and thank the reviewer for his/her suggestions. We deepen the description of the aetiology of the NTFH group. As for NTFH group, we excluded patients reporting aetiology with alcohol abuse and autoimmune diseases to represent a more homogenous class. We inserted these details and also a scientific diagram showing the weight-bearing areas as required by the other reviewer. Reviewer 1 can find these changes on page 3 from line 96 to line 97 and from line 102 to 103.

  • Did the multi-organ donors suffer from any musculoskeletal disease? This is to mention in Materials and Methods.

-) We thank the reviewer for his/her suggestions. Multi-organ donors did not suffer from any musculoskeletal disease. We selected the donors through the bone bank program for tissue donation after the family’ s donor consent. Femur harvesting was performed within six hours from asystole and involved its excision and placement in Dulbecco-modified Eagle medium with L-glutamine, NaHCO3, and antibiotics, and storage at 4°C. Reviewer 1 can find these changes on page 3 from line 109 to line 112.

  • It would be of interest some systematization of the manuscript. For example, a short presentation and reasoning of the markers studied, in the section Introduction.

-) We agree and thank the reviewer for his/her suggestions. We inserted a brief presentation of markers used in the manuscript by adding these details in the introduction section. We entered further references in the document. Reviewer 1 can find these changes on page 2 from line 71 to line 83.

  • At the description of immunohistochemical analysis, the authors mention: “The measurement of positive cells and area for each marker was done on the entire osteochondral sample (10× objective lens) and expressed as a percentage of positive cells and area on a scale from 0 (no protein expression) to 100 (the highest protein expression).” How many fields did they measure in average for each sample? Please specify.

-) We agree and thank the reviewer for his/her suggestions. We added the lacking information in the document. Six microscopic fields (100 x magnification) were assessed for each sample with a semi-quantitative method by an investigator (IB). We fragmented the cartilage and bone by testing the positivity for each marker.  Reviewer 1 can find these changes on page 4 from line 136 to line 140. We changed figures by providing high magnification pictures in all figures, as suggested by reviewer 2.

  • There is no description, how the staining intensity was assessed in cartilage and subchondral bone. Did they select zones apart from the tidemark, if this was fragmented? Please specify this.   

-) We agree and thank the reviewer for his/her suggestions. We firstly segmented cartilage and the subchondral bone by selecting zones apart from the tidemark, especially for PTFH group where the tidemark was fragmented. Then, we defined the threshold positivity using the Hue/Saturation/Intensity system with the NIS-Element software. Reviewer 1 can find these details on page 4 from line 136 to line 146.

Reviewer 2 Report

This study addresses a very interesting and topical issue of bone-cartilage cross-talk. It has direct relevance to future treatment developments for femoral head osteonecrosis, as well as for hip osteoarhtiritis. In the present study, femoral head osteonecrosis patients were divided into two groups of different aetiologies: non-traumatic (NTFH) and post traumatic (PTNFH). The authors need to clarify what ‘N’ stands for in the PTNFH acronym. They also used a small but very valuable group of non-diseased controls.

My major comments are specified below:

  1. Taking of the biopsies. The authors mentioned that the biopsies were taken from the weight-bearing area, but they need to be much more precise and relate to femoral head anatomy, for example by referring to anterior, posterior, medial, etc regions. A diagram in a supplementary material would be very useful.
  2. Decalcification: the authors have used a very harsh method. Normally for immunohistochemistry, a milder EDTA-based method is used. The authors should provide any references for the previous success of using hydrochloric/formic acid for decalcification followed by staining with the chosen antibodies.
  3. Scoring positive cells/area following antibody staining. The method description is not clear enough - whether it was positive cells or positive areas measured. If this depended on the antibody used, this should be clearly stated and the counter-stain for cell nuclei (for total cells) should be mentioned. Was it also done by two independent observers? The data consistency between the observers should be prvided, ideally as a supplementary table.
  4. All figures/graphpad graphs: because of a relatively low numbers of samples tested, individual donors’ results should be incorporated into the graphs on top of the existing bar/error bars. There is such an option on Graphpad, and this will significantly improve reader’s confidence in the presented data.
  5. The importance of testing Autaxin should be stated in the Introduction.
  6. All IH figures should have higher-magnification images, and the y axes on the graphs should indicate whether they present % of positive cells or % area. Currently, it is not clear whether the markers were mostly associated with the cells, with ECM in the pericellular matrix, or distributed fairly similar across the whole ECM.
  7. Higher-magnification histology images would be also useful to assess the presence of osteoblasts and osteoclasts, indicating active bone remodelling process at the site of interest (some images appear to contain cuboid osteoblasts).
  8. The authors should relate more to similar histological features and cellular responses in OA bone marrow lesions (BMLs); this would help to broaden their discussion on the importance of bone-cartilage cross-talk in both pathologies.

Author Response

Responses to reviewer 2

We thank Reviewer 2 and followed his/ her suggestions to improve the quality of the manuscript.

  1. Taking of the biopsies. The authors mentioned that the biopsies were taken from the weight-bearing area, but they need to be much more precise and relate to femoral head anatomy, for example by referring to anterior, posterior, medial, etc regions. A diagram in a supplementary material would be very useful.

-) We thank the reviewer for his/her kind suggestions. We inserted a scientific diagram reporting the weight-bearing areas of the femoral epiphysis where we harvested biopsies (see Supplementary Figure 1).

  1. Decalcification: the authors have used a very harsh method. Normally for immunohistochemistry, a milder EDTA-based method is used. The authors should provide any references for the previous success of using hydrochloric/formic acid for decalcification followed by staining with the chosen antibodies.

-) We thank the reviewer for his/her kind comments. We agree with the Reviewer that EDTA-based method is a milder approach than using hydrocloridric and formic acid solution. However, we often adopted sample processing in our lab using this solution for both histology and immunohistochemistry. We inserted our previous reference (Reference 42) reporting immunohistochemistry for several antigens on osteochondral samples decalcified with this solution. Reviewer 2 can find these changes on Page 3, Line 115. 

  1. Scoring positive cells/area following antibody staining. The method description is not clear enough - whether it was positive cells or positive areas measured. If this depended on the antibody used, this should be clearly stated and the counter-stain for cell nuclei (for total cells) should be mentioned. Was it also done by two independent observers? The data consistency between the observers should be provided, ideally as a supplementary table.

-) We thank the reviewer for his/her kind suggestions. We clarified these aspects in the text as kindly suggested by the reviewer. In general, we found positive staining for areas only for collagen type I in the subchondral bone, whereas all other markers showed marked cell positivity. We reported such details in each graph in the Y-axis, as suggested by the reviewer. All cell nuclei were counter-stained with CAT haematoxylin as already stated in the text. An observer examined all tissue samples by giving a semi-quantification of each marker using a NIS software for image analysis. In particular, six microscopic fields (100 x magnification) were assessed for each specimen. We followed the reviewer’s suggestion to increase the data consistency by adopting the scatter plot diagram reporting data from each sample. Reviewer 2 can find these changes on Figures 1b, 2b, 3b, 3b, 4b and 5b and on Page 4, Line 136-140. 

  1. All figures/graphpad graphs: because of a relatively low numbers of samples tested, individual donors’ results should be incorporated into the graphs on top of the existing bar/error bars. There is such an option on Graphpad, and this will significantly improve reader’s confidence in the presented data.

-) We agree and thank the reviewer for his/her kind suggestions. We modified GraphPad graphs, as suggested by reviewer 2 to improve the reader’s confidence in the presented data. Reviewer 2 can find these changes on Figures 1b, 2b, 3b, 3b, 4b and 5b.

  1. The importance of testing Autaxin should be stated in the Introduction.

-) We agree and thank the reviewer for his/her kind suggestions. We inserted a brief description of autotaxin and other markers tested in the introduction section, as also suggested by reviewer 1. Reviewer 2 can find these changes on Page 2, Line 71-83. 

  1. All IH figures should have higher-magnification images, and the y axes on the graphs should indicate whether they present % of positive cells or % area. Currently, it is not clear whether the markers were mostly associated with the cells, with ECM in the pericellular matrix, or distributed fairly similar across the whole ECM.

-) We agree and thank the reviewer for his/her kind suggestions. We changed the graphs and the y-axis by reporting % positive cells or % positive area. To provide better visualization of positive signals, we changed all figures by inserting pictures with high magnification as suggested by the reviewer. Reviewer 2 can find these changes in Figures 2a, 2b, 3a, 3b, 4a, 4b, 5a, 5b.

  1. Higher-magnification histology images would be also useful to assess the presence of osteoblasts and osteoclasts, indicating active bone remodelling process at the site of interest (some images appear to contain cuboid osteoblasts).

-) We thank the reviewer for his/her kind suggestions. We inserted high magnification pictures, as suggested by the reviewer to appreciate the regions of interest. Reviewer 2 can find these changes in Figure 1 a.

  1. The authors should relate more to similar histological features and cellular responses in OA bone marrow lesions (BMLs); this would help to broaden their discussion on the importance of bone-cartilage cross-talk in both pathologies.

-) We appreciate the reviewer’s comment and exploited his suggestion to broaden the discussion to highlight the biological relevance of bone-cartilage cross-talk. Reviewer 2 can find these changes on Page 12, from line 291 to line 307. 

Round 2

Reviewer 1 Report

My only comment refers to the abbreviation PTFH. If the authors use PTFH instead of PTNFH, then the graph labels in Figure 1,2,3,5 should also be harmonized with this abbreviation.

The authors performed a thorough major review, and I think the manuscript now fulfills all citeria to be published.

Author Response

Responses to reviewer 1

We thank Reviewer 1 and followed his/ her suggestions to improve the quality of the manuscript and ensure a better comprehension from the readers.

My only comment refers to the abbreviation PTFH. If the authots use PTFH instead of PTNFH, then the graph labels in Figure 1, 2, 3, 5 should also be harmonized with this abbreviation.

-) We thank the Reviewer for his/her comments. We had already harmonized all figures and tables with the proposed acronym PTFH. As kindly suggested by the Reviewer, we checked all figures, tables and text. We found a mistake in the text on Page 4 in Line 164.  We apologise for the mistake; we corrected it by using the new abbreviation PTFH.

-) We corrected some spelling errors within the text and the title.

Reviewer 2 Report

The authors have adequately addressed my comments. Suppl figure legend is not available in the submission

Author Response

Responses to reviewer 2

We thank Reviewer 2 and followed his/ her suggestions to improve the quality of the manuscript and ensure a better comprehension from the readers.

The authors have adequately addressed my comments. Suppl figure legend is not available in the submission.

-) We thank the Reviewer for his/her comments. We apologize for the mistake. We inserted the figure Legend of the supplementary figure (See Supplementary Figure 1).

-) We corrected some spelling errors within the text and the title.
